# PLCβ2 negatively regulates the inflammatory response to virus infection by inhibiting phosphoinositide-mediated activation of TAK1

Lin Wang[1], Yilong Zhou[1], Zijuan Chen[2], Lei Sun[3], Juehui Wu[1], Haohao Li[1], Feng Liu[1], Fei Wang[1], Chunfu Yang[4], Juhao Yang[4], Qibin Leng[4], Qingli Zhang[5], Ajing Xu[5], Lisong Shen[5], Jinqiao Sun[6], Dianqing Wu[7], Caiyun Fang[8], Haojie Lu[8], Dapeng Yan[2] & Baoxue Ge[1]

Excessive or uncontrolled release of proinflammatory cytokines caused by severe viral infections often results in host tissue injury or even death. Phospholipase C (PLC)s degrade phosphatidylinositol-4, 5-bisphosphate (PI(4,5)P2) lipids and regulate multiple cellular events. Here, we report that PLCβ2 inhibits the virus-induced expression of pro-inflammatory cytokines by interacting with and inhibiting transforming growth factor-β-activated kinase 1 (TAK1) activation. Mechanistically, PI(4,5)P2 lipids directly interact with TAK1 at W241 and N245, and promote its activation. Impairing of PI(4,5)P2's binding affinity or mutation of PIP2-binding sites on TAK1 abolish its activation and the subsequent production of pro-inflammatory cytokines. Moreover, PLCβ2-deficient mice exhibit increased expression of proinflammatory cytokines and a higher frequency of death in response to virus infection, while the PLCβ2 activator, m-3M3FBS, protects mice from severe Coxsackie virus A 16 (CVA16) infection. Thus, our findings suggest that PLCβ2 negatively regulates virus-induced pro-inflammatory responses by inhibiting phosphoinositide-mediated activation of TAK1.

[1] Shanghai Pulmonary Hospital, Tongji University School of Medicine, 200433 Shanghai, China. [2] Department of Immunology, School of Basic Medical Sciences & Shanghai Public Health Clinical Center, Key Laboratory of Medical Molecular Virology of MOE/MOH, Fudan University, 200032 Shanghai, China. [3] School of Pharmacy, Shanghai Jiao Tong University, 200240 Shanghai, China. [4] Institut Pasteur of Shanghai, 200031 Shanghai, China. [5] Xinhua Hospital Affiliated to Shanghai Jiao Tong University School of Medicine, 200000 Shanghai, China. [6] Department of Clinical Immunology, Children's Hospital of Fudan University, 201102 Shanghai, China. [7] Department of Pharmacology, Yale School of Medicine, New Haven, CT 06520, USA. [8] Department of Chemistry and Institutes of Biomedical Sciences, Fudan University, 200032 Shanghai, China. These authors contributed equally: Lin Wang, Yilong Zhou, Zijuan Chen. Correspondence and requests for materials should be addressed to H.L. (email: luhaojie@fudan.edu.cn) or to D.Y. (email: dapengyan@fudan.edu.cn) or to B.G. (email: gebaoxue@sibs.ac.cn)

nfectious diseases, especially viral infections, remain a serious threat to humanity. Both clinical and experimental studies have found a correlation between the excessive burst in proinflammatory cytokines, known as a cytokine storm, and the morbidity and mortality associated with infectious diseases, such as influenza pneumonia, hand, foot, and mouse disease (HFMD), and bacterial sepsis[1–5]. Inflammatory mediators induced during a severe viral infection usually include interferons, tumor necrosis factors, interleukins, and chemokines. More than 150 cytokines have been proposed to contribute to the development of a cytokine storm, which, in combination with a knowledge of the relevant cytokine/chemokine signaling, would be novel and favorable antiviral and autoimmune therapeutic targets[3]. However, the precise mechanism of induction of cytokine storm is largely unknown.

The production of proinflammatory cytokines is an important part of innate immunity. Innate immunity is an evolutionarily conserved defense mechanism against microbial pathogens and is essential for the activation of the adaptive immune response[6,7]. Most viruses produce double-stranded RNA (dsRNA) during the infection. dsRNA or its mimic, polyinosine/polycytosine (poly (I: C)), stimulates TLR3 and leads to a cascade of downstream signaling that ultimately activate IFN regulatory factor 3 (IRF3) and nuclear factor κB (NF-κB), resulting in the expression of type I IFNs and proinflammatory cytokines, such as TNF, IL-6, and IL-12, respectively[8–10].

Transforming growth factor-β-activated kinase 1 (TAK1) is a member of the mitogen-activated protein kinase kinase kinase (MAPKKK) family and was found to be central to the activation of the NF-κB, c-Jun N-terminal kinase (JNK), and p38 pathways[11,12]. TAK1 forms a complex with TRAF6 and associated proteins, including TAB1 (TAK1 activator) and TAB2/3 (ubiquitin-binding proteins)[13,14]. TRAF6 is a RING domain ubiquitin ligase that functions with Ubc13 and Uev1A to catalyze the formation of Lys-63 (K63)-linked polyubiquitin of TAK1. TAB2 and TAB3 contain a highly conserved zinc finger motif termed NZF, which binds to K63-linked polyubiquitin chains of TRAF6 and results in the activation of TAK1[11,12]. Ubiquitin-activated TAK1 then phosphorylates MKKs, leading to the activation of the JNK and p38 kinase pathways. On the other hand, several different phosphatases, including PP2C and PP6, have been identified as negative regulators of TAK1 by dephosphorylating it at Thr187[15]. TAK1 activity can also be downregulated by de-ubiquitinases. For example, Cyld and USP4 cleave K63-linked polyubiquitin chains bound to TAK1, whereas Itch targets TAK1 at K48-linked ubiquitin chains and sends it to degradation[16,17]. However, it remains unclear how TAK1 activation is resolved during virus infections.

Phospholipase C (PLC)s are key signaling proteins in response to many hormones, neurotransmitters, growth factors, and other extracellular stimuli. PLCs degrade phosphatidylinositol-4, 5-bisphosphate (PI(4,5)P2) to generate diacylglycerol (DAG) and inositol 1,4,5-trisphosphate (IP3), two important second messengers for protein kinase C (PKC) activation and intracellular calcium release from the endoplasmic reticulum, respectively[18]. Of the PLCβ isoforms, PLCβ1 and PLCβ3 exhibit wide tissue distribution, whereas PLCβ2 is principally expressed in hematopoietic cells and mediates chemoattractant-induced production of superoxide and activation of protein kinases[18–20]. Recently, it has been shown that suppression of PLCβ2 by LPS plays a role in switching M1 macrophages into an M2-like state or downregulating chemokine receptor signaling in B cells[21–24]. However, the role of PLCβ2 in the regulation of antiviral innate immune responses remains unknown.

In the present study, we discovered a role for PLCβ2 in controlling antiviral inflammatory responses. We obtained molecular and cellular evidence that PLCβ2 down-regulates virus-induced activation of TAK1, as well as the subsequent production of proinflammatory cytokines through the degradation of PIP2. Our findings indicate that PLCβ2 is a negative regulator of the virus-induced inflammatory response and treatment with a PLC activator could serve as a new therapeutic strategy for viral infections.

## Results

**RNA virus infection specifically induces PLCβ2 expression.** HFMDs caused by enteroviruses, such as CVA16 and EV71, are a group of infectious disease that affects millions of people globally, especially children under the age of 5[25]. Severe HFMDs are often associated with aseptic meningitis, brain stem encephalitis, and acute flaccid paralysis and can be fatal[26,27]. To determine which molecules were important in HFMD, we collected blood samples from five healthy controls and six patients with clinically diagnosed HFMDs. After treated with red blood cell lysis, the white blood cells of each group patient were mixed and analyzed by mass spectrometry. Our results revealed that PLCβ2 is much highly expressed in patients compared to healthy controls (Fig. 1a). We analyzed the protein level of PLCβ2 by western blotting. As shown in Fig. 1b, the protein abundance of PLCβ2 was also significantly higher in HFMD patients compared with controls, which is consistent with our previous mass spectrometry results.

We next determined whether PLCβ2 is induced by CVA16 infection. Indeed, both CVA16 and poly (I:C), a viral dsRNA mimic, increased PLCβ2 protein abundance in macrophages (Fig. 1c, d). Similarly, two other RNA viruses, influenza A virus H5N1 and vesicular stomatitis virus (VSV), also increased PLCβ2 protein levels in mouse peritoneal macrophages (Supplementary Fig. 1a, b). However, PLCβ2 protein levels were not changed in macrophages stimulated with herpes simplex virus (HSV), a DNA virus, or when stimulated with LPS or PGN, two bacterial-derived stimuli (Supplementary Fig. 1c–e). Bacillus Calmette-Guérin (BCG) also did not regulate PLCβ2 protein levels (Supplementary Fig. 1f). These results indicate that induction of PLCβ2 may be a general phenomenon that occurs after RNA virus infection.

To determine which signaling pathway is involved in PLCβ2 expression, we further examined the poly(I:C)-induced upregulation of PLCβ2 in peritoneal macrophages derived from $Tlr3^{-/-}$ mice or wild-type macrophages pre-treated with PDTC (a NF-κB inhibitor), PD98059 (a MEK inhibitor), SP600125 (a JNK inhibitor), or SB203580 (a p38 inhibitor). Poly(I:C)-induced upregulation of PLCβ2 was attenuated in $Tlr3^{-/-}$ macrophages and in wild-type macrophages treated with SB203580 (Fig. 1e, f and Supplementary Fig. 1g–i). These results suggest that PLCβ2 induction is stimulated by dsRNA through the TLR3-p38 pathway.

**PLCβ2 suppresses RNA virus-induced inflammation.** As a number of clinical studies have suggested that the massive induction of proinflammatory cytokines, such as IL-6 could be responsible for severity of pathogenesis in HFMD[26,28–31], we investigated the role of PLCβ2 in CVA16-induced production of proinflammatory cytokines. Two-week-old wild-type or $Plcb2^{-/-}$ mice were infected with CVA16 virus for 5 days, then expression levels of proinflammatory cytokines were analyzed via quantitative RT-PCR analysis. Skeletal muscle tissue from $Plcb2^{-/-}$ mice had significantly higher mRNA levels of Tnf (Fig. 2a), Il6 (Fig. 2b), and Il12 (Fig. 2c) compared with wild-type mice, suggesting that PLCβ2 negatively regulates virus-induced expression of proinflammatory cytokines in vivo. Additionally, $Plcb2^{-/-}$ mice produced more Tnf, Il6, and Il12 than wild-type mice after challenge with VSV (Supplementary Fig. 2a–c). Moreover, when

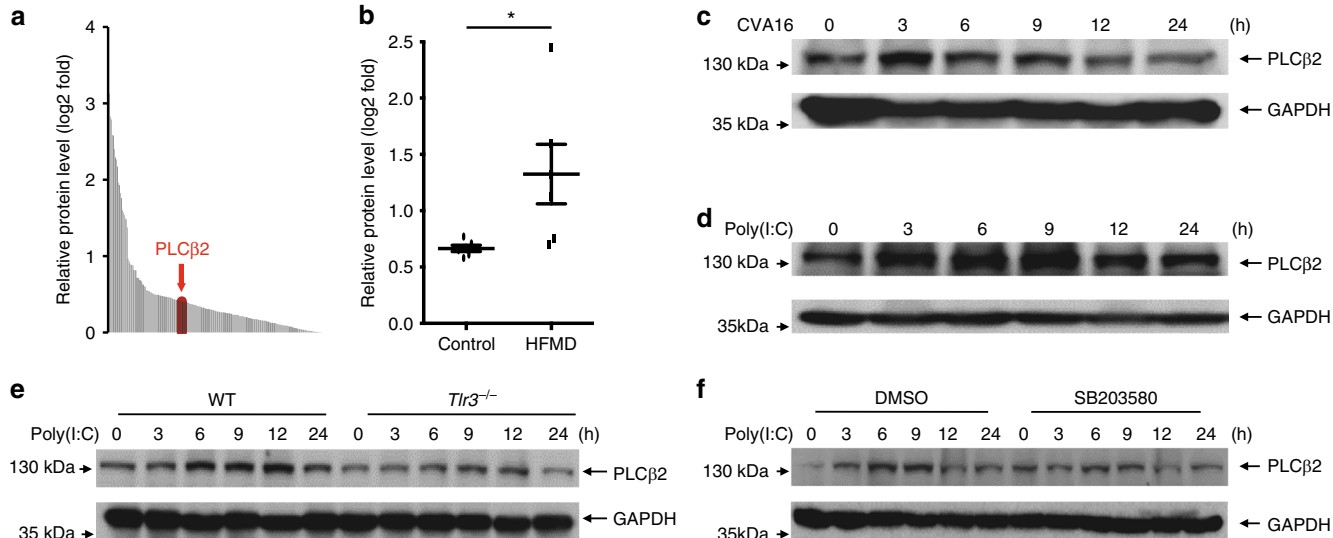

**Fig. 1** RNA virus infection specifically induces PLCβ2 expression. **a** Protein expression ratio of white blood cells mixture from control and HFMD group patients detected by mass spectrometry. **b** Immunoblot (IB) and analysis of PLCβ2 in PBMC from control and HFMD patients. **c, d** IB of PLCβ2 in lysates of mouse peritoneal macrophages infected with CVA16 virus (**c**) or stimulated with poly(I:C) (**d**) for the indicated times. **e** IB of PLCβ2 in wild-type or $Tlr3^{-/-}$ mouse peritoneal macrophages stimulated with poly(I:C) for the indicated times. **f** IB of PLCβ2 in mouse peritoneal macrophages pretreated for 1 h with the p38 inhibitor SB203580 before poly(I:C) stimulation for the indicated times. *$p < 0.05$ by unpaired $t$-test (**b**). Data are representative of three experiments with at least three independent biological replicates (mean and s.e.m. of $n = 6$ per group in **b** and $n = 3$ cultures in **c–f**)

stimulated with poly(I:C) or TLR7 ligand r-848, *Plcb2*-deficient or knockdown macrophages exhibited enhanced production of the proinflammatory cytokines compared with wild-type macrophages (Fig. 2d, e and Supplementary Fig. 2d), while poly(I:C) transfection of macrophages showed comparable *Il6* expression between wild-type and *Plcb2* knockdown macrophages (Supplementary Fig. 2d).

We next stimulated primary macrophages derived from *Plcb2*-deficient or wild-type mice with poly(I:C) and analyzed the activation of the MAP kinase and NF-κB pathways. When stimulated with poly(I:C), *Plcb2*-deficient macrophages exhibited greater activation of the MAP kinases and NF-κB pathways compared with wild-type macrophages (Fig. 2f), which exhibited no difference after stimulation with HSV, LPS, PGN, or other bacterial pathogens (Supplementary Fig. 2e–i). These data suggest that PLCβ2 specifically suppresses RNA virus-mediated production of proinflammatory cytokines by downregulating the MAPK and NF-κB pathways.

To further explore the potential redundant functions of other PLCβ isoforms in TLR3 signal in macrophages. We detected the *Plcβ* family expression in mice peritoneal macrophages and found *Plcb2* had the most abundant mRNA level among *Plcβ1–4* (Fig. 2g). Moreover, we knocked down *Plcβ1*, *Plcβ2*, *Plcβ3* and *Plcβ4* in mice macrophages and found that only *Plcβ2* knockdown promoted poly(I:C)-induced *Tnf* and *Il6* expression (Fig. 2h, i). Taken together, these functional and mechanical studies indicated that PLCβ2 but not other PLCβ proteins modulate TAK1 activation in TLR3 pathway.

**PLCβ2 interacts with TAK1.** Viral RNA induces proinflammatory cytokine production via a signaling complex composed of TRAF6, TRAF2, TAK1, TAB1, and p38[12]. To further investigate the mechanism underlying the modulation of inflammation by PLCβ2, we assessed the interaction between PLCβ2 and TRAF6, TRAF2, TAK1, TAB1 and p38. Only TAK1 was found to interact with PLCβ2 in HEK293T cells (Fig. 3a–c). The Plcc and C2_2 domain of PLCβ2 are responsible for the

interaction (Supplementary Fig. 3a, b). Using purified recombinant TAK1 proteins in an in vitro glutathione S-transferase (GST) precipitation assay, we found that TAK1 proteins could associate with purified recombinant His-PLCβ2 or endogenous PLCβ2 from RAW264.7 cells (Fig. 3d, e). Furthermore, stimulation with poly (I:C) enhanced the interaction and co-localization of endogenous TAK1 with endogenous PLCβ2 in mouse peritoneal macrophages (Fig. 3f, g), suggesting that the interaction of TAK1 with PLCβ2 was constitutive, which was affected by poly(I:C) stimulation. Moreover, our confocal microscopy showed that a part of TAK1 co-localized with EEA1 (early endosome marker) after poly (I:C) treatment (Supplementary Fig. 3c), which indicated that endosomal PLCβ2 was specifically associated with endosomal TAK1 activation in the TLR3 pathway. Lastly, among PLCβ1–3, only PLCβ2 was associated with TAK1 (Fig. 3h).

**PLCβ2 inhibits TAK1 activation.** As PLCβ2 interacts with TAK1, we next investigated whether PLCβ2 has any effect on TAK1 activation. The association of TAB1 with TAK1 induces TAK1 activity[13]. Indeed, co-expression of TAK1 with TAB1 greatly induced TAK1 phosphorylation, but PLCβ2 inhibited TAK1 phosphorylation and TAK1–TAB1 interaction (Fig. 4a and Supplementary Fig. 4a). When activated, TAK1 phosphorylates IKKβ or MKKs and leads to the activation of the NF-κB, JNK, and p38-signaling pathways[32]. Co-expression of TAK1 and TAB1 in HEK293T cells substantially increased NF-κB and AP-1 reporter activities, whereas the expression of PLCβ2 inhibited this increase in a dose-dependent manner (Fig. 4b, c). It is known that the activation of TAK1 depends on its K63-linked polyubiquitination. Expression of TAB1 enhanced the K63-linked polyubiquitination of TAK1, which was dramatically decreased by PLCβ2 over-expression (Supplementary Fig. 4b). Similarly, poly(I:C) induced an increased phosphorylation of TAK1 at Thr187 and Ser192 in primary macrophages derived from $Plcb2^{-/-}$ mice compared with those derived from wild-type mice (Fig. 4d). Moreover, 5Z-7-OZ, a selective inhibitor of TAK1, markedly decreased poly (I:C)-induced activation of MAPKs and

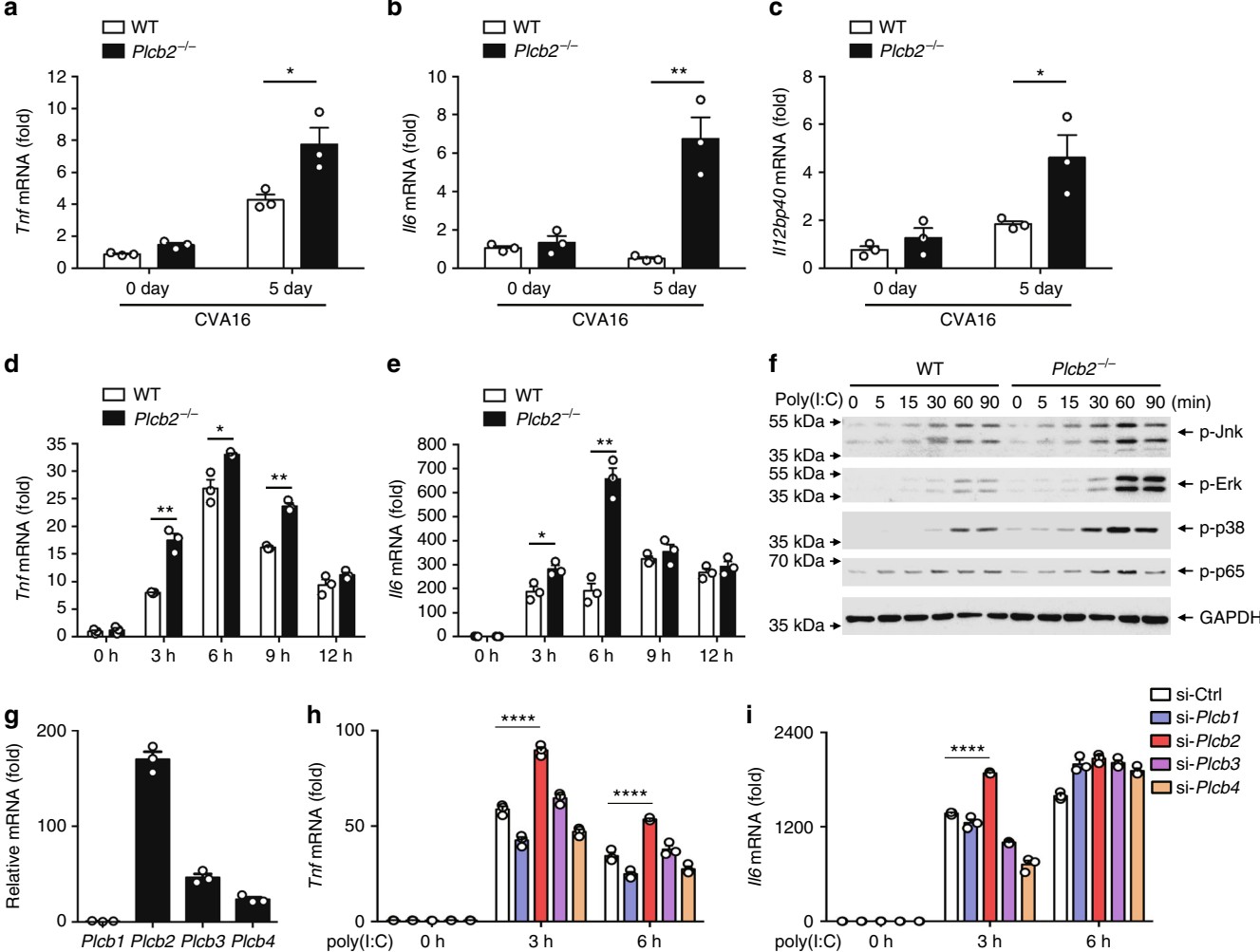

**Fig. 2** PLCβ2 suppresses RNA virus-induced inflammation. **a–c** Q-PCR analysis of relative *Tnf* (**a**), *Il6* (**b**), and *Il12p40* (**c**) mRNA levels in skeletal muscle tissue from 14-day-old wild-type or *Plcb2⁻/⁻* mice infected with 50 μl of CVA16 virus (1.5 × 10⁴ PFU/mouse) for 0 or 5 days. **d, e** Q-PCR analysis of relative *Tnf* (**d**) and *Il6* (**e**) mRNA in wild-type or *Plcb2⁻/⁻* macrophages treated with poly(I:C) for the indicated times. **f** IB of lysates from wild-type or *Plcb2⁻/⁻* macrophages treated with poly(I:C) for the indicated times using anti-phospho-antibodies. **g** Q-PCR analysis of relative *Plcb1-4* mRNA levels in mice peritoneal macrophages. **h** Q-PCR analysis of relative *Tnf* (**h**) and *Il6* (**i**) mRNA in *Plcb1-4* silence macrophages stimulated with poly(I:C) for the indicated times. *$p < 0.05$; **$p < 0.01$ and ****$p < 0.0001$ by unpaired *t*-test (**a–e**, **h** and **i**). Data are representative of three experiments with at least three independent biological replicates (mean and s.e.m. of $n = 3$ cultures in **d–i** or $n = 3$ mice per group in **a–c**)

also eliminated the enhancement effects of PLCβ2 deficiency on poly (I:C)-induced MAPK pathway activation (Fig. 4e). These results suggest that the negative regulation of inflammation by PLCβ2 is largely dependent on suppressing the activation of TAK1.

To further identify the clinical correlation of PLCβ2 and TAK1 activation, we performed an array analysis of PLCβ2 expression and p-TAK1 in blood samples from 30 HFMDs patients; we found an inverse correlation between TAK1 activation and PLCβ2 protein levels (Fig. 4f). The first representative 12 samples were divided into two groups according to the level of TAK1 phosphorylation. The results indicated that samples from the group with higher TAK1 phosphorylation also had lower PLCβ2 expression as compared to the group with lower TAK1 phosphorylation (Fig. 4g). Furthermore, IL-6 protein abundance was significantly higher in HFMD patients with higher TAK1 phosphorylation (Supplementary Fig. 4c). These results suggest that PLCβ2 may suppress the activation of TAK1, thus reducing the production of IL-6 in patients with HFMD.

**PLCβ2 inhibits TAK1 activation via PIP2**. Upon activation, PLC hydrolyzes PIP2 into IP3 and DAG. IP3 mediates the release of Ca²⁺ from intracellular stores. DAG activates PKC to regulate localized signaling[18–20]. According to structure of PLCβ2 histidine located on 327 and 374 are important for the catalytic activity, thus we mutated H327 and H374 to alanine to generate phospholipase-inactive mutant of PLCβ2[33]. Expression of wild-type PLCβ2 inhibited TAK1-induced activation of NF-κB and AP-1 reporter gene, but the phospholipase-inactive mutant did not (Fig. 5a, b). Consistent with this, wild-type PLCβ2 inhibited TAB1 and TAK1 interaction, as well as TAB1-mediated TAK1 phosphorylation, while mutation of phospholipase-active site abolished PLCβ2's inhibition on TAB1 and TAK1 interaction and TAK1 phosphorylation (Fig. 5c, Supplementary Fig. 5a), although phospholipase-inactivate mutant could still interact with TAK1 (Supplementary Fig. 5b). We have examined those PLCβ2 mutants as shown in Supplementary Fig. 3a and b, if they have any effect on the activation of TAK1. The data showed that PLCβ2 catalytic domain-Plcc domain was crucial for PLCβ2's inhibition to TAK1-induced NF-κB and AP-1 signal, while

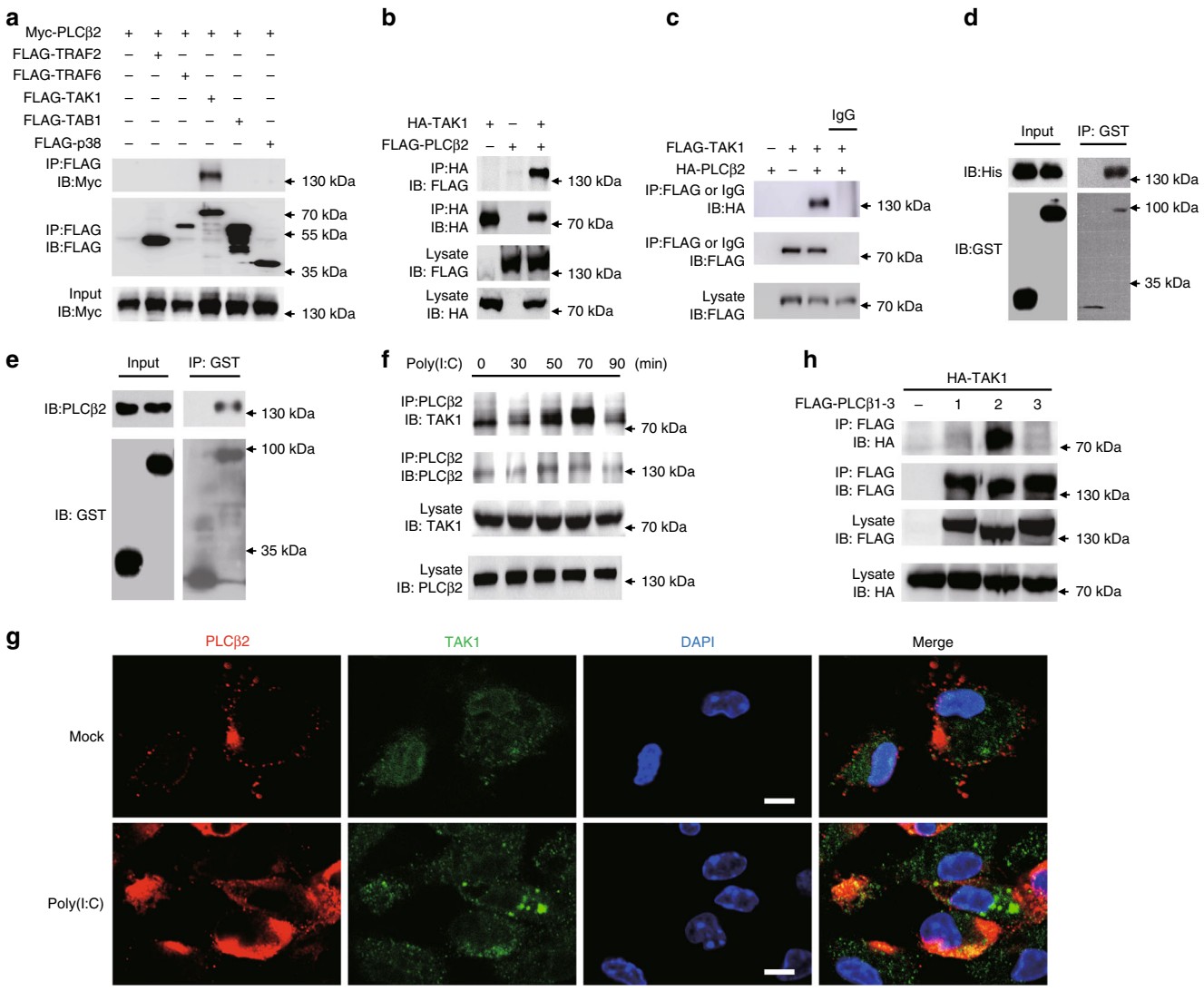

**Fig. 3** PLCβ2 interacts with TAK1. **a–c** IB or immunoprecipitates (IP) of lysates from HEK293T cells transfected with various plasmids as indicated. **d** In vitro GST precipitation assay of TAK1 and PLCβ2. **e** Precipitation of endogenous PLCβ2 from RAW264.7 cells expressing GST or GST-TAK1. **f** Endogenous interaction of PLCβ2 with TAK1 in poly(I:C)-treated primary macrophages. **g** Confocal analysis of colocolization of TAK1 and PLCβ2 in peritoneal macrophages stimulated with poly(I:C) for 4 h. Scale bar, 5 μm. **h** IB or IP of lysates from HEK293T cells transfected with FLAG-tagged PLCβ1-3 and HA-tagged TAK1. Data are representative of three experiments with at least three independent biological replicates (n = 3 cultures in **f**, **g**)

PLCβ2 mutants that do not interact with TAK1 have no significant inhibitory effect on the TAK1-mediated activation of NF-kappaB or AP-1 reporter gene (Supplementary Fig. 5c, d). These results suggest that direct interaction of PLCβ2 with TAK1 is prerequisite, but not sufficient for the regulation of TAK1 activity. Thus, we then hypothesized that the PLCβ2 substrate PIP2 may be responsible for TAK1 activity modulation. In order to test this hypothesis, we detected interaction of TAK1 with TAB1 with carrier-mediated transfer of exogenous PI(4,5)P2 lipids and found PI(4,5)P2 could promote TAK1–TAB1 complex formation (Supplementary Fig. 5e). Consistently, poly(I:C) induced a more phosphorylation of TAK1 when PI(4,5)P2 was transferred into mice peritoneal macrophages (Fig. 5d), as well as activation of downstream MAPKs and NF-κB (Fig. 5e). Conversely, neomycin, an aminoglycoside antibiotic that binds to PIP2 with high affinity leading to separation of phospholipid-binding proteins from phospholipids, markedly impaired poly(I:C)-induced TAK1 phosphorylation (Fig. 5f), as well as the activation of the MAPK and NF-κB pathways (Fig. 5g). Moreover, poly (I:C)-induced *Tnf* and *Il6* mRNA levels were also decreased with neomycin

treatment (Fig. 5h, i). Additionally, neomycin eliminated the enhancement effects of PLCβ2 deficiency on poly (I:C)-induced MAPKs and NF-κB pathway activation (Fig. 5j), as well as *Tnf* and *Il6* mRNA production (Fig. 5k, l). These results suggest that negative regulation of inflammation by PLCβ2 may be dependent on PIP2 hydrolysis.

To further investigate the mechanisms by which PIP2 activates TAK1, we first determined whether PIP2 interacts with TAK1. Using a dot blot assay of PIP2 and purified TAK1, we found that there was a direct interaction between TAK1 and PI (4, 5)P2 (Fig. 5m). We then made a prediction using the http://molsoft.com/~eugene/moda/modamain.cgi website with TAK1 structure (PDB code: 2eva) to identify the sites on TAK1 that are required to bind to PIP2. The prediction worked out with membrane optimal docking area (MODA) values, which indicates how likely a particular residue is to interact with membrane. The higher MODA values means higher probability, generally from which above 50 means much likely interface residues. We chose amino acid sites with a MODA > 400, including $G^{234}$ (MODA = 493.32), $P^{235}$ (MODA =

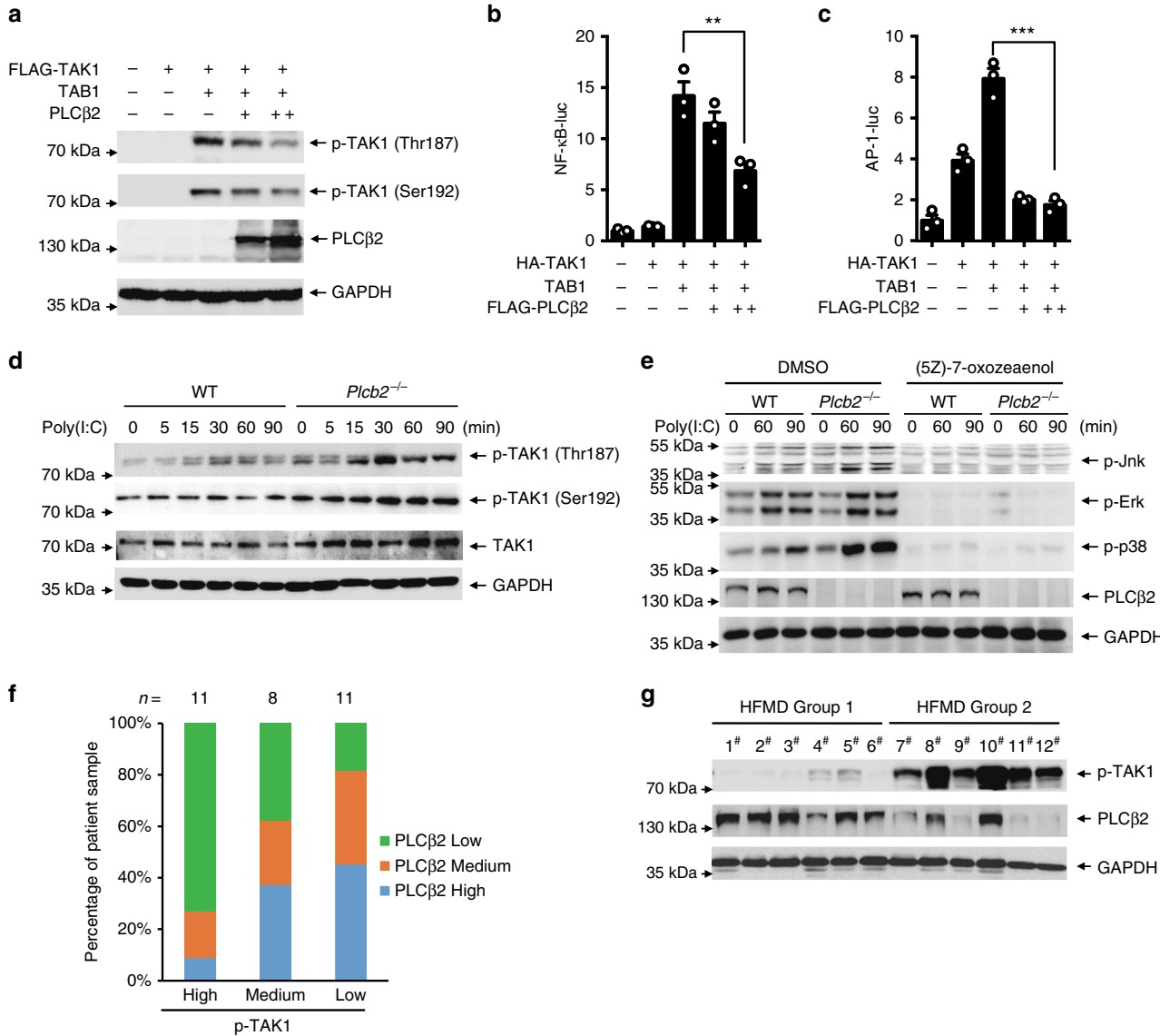

**Fig. 4** PLCβ2 inhibits TAK1 activation. **a** IB of cell lysates from HEK293 cells expressing TAK1 and TAB1 in the presence or absence of PLCβ2. **b**, **c** Reporter assay of NF-κB (**b**) and AP-1 (**c**) activation in HEK293T cells expressing TAK1 and TAB1 with increasing concentrations of PLCβ2. **d** Immunoblot analysis of lysates from wild-type or $Plcb2^{-/-}$ macrophages treated with poly(I:C) for the indicated times using phospho-TAK1 antibody. **e** IB of lysates from wild-type or $Plcb2^{-/-}$ macrophages pretreated with TAK1 inhibitor, (5Z)-7-oxozeaenol, for 1 h, then stimulated with poly(I:C) for the indicated times using anti-phospho-antibodies. **f**, **g** IB of PLCβ2 and p-TAK1 in PBMC from HFMD patients. Both PLCβ2 and p-TAK1 levels were classified as low, medium, or high based on the intensities of each band; the percentages of patients classified in each category are depicted the histogram in (**f**) and the representative IB is shown in (**g**). **p < 0.01 and ***p < 0.001 by unpaired t-test (**b**, **c**). Data are representative of three experiments with at least three independent biological replicates (n = 3 cultures in **d**, **e**)

422.43), $F^{237}$ (MODA = 1298.01), $R^{238}$ (MODA = 587.27), $W^{241}$ (MODA = 1783.65) and $N^{245}$ (MODA = 400.71), then mutated these amino acids to alanine. As a result, dot assays of PIP2 with purified TAK1 and its mutants showed that $W^{241}$ and $N^{245}$ are crucial for TAK1 binding to PIP2 (Fig. 5n). To characterize the role of W241 and N245 on TAK1 activation, we complemented $TAK1^{-/-}$ A549 cells with wild-type TAK1 or TAK1 ($W^{241}$A) (M1) mutants (Supplementary Fig. 5f, g). With poly (I:C) stimulation, phosphorylation of TAK1 and MAPKs were dramatically decreased in $TAK1^{-/-}$ A549 cells transfected with TAK1 (M1) compared to cells transfected with wild-type TAK1 (Fig. 5o). In addition, $TAK1^{-/-}$ A549 cells complemented with TAK1 ($W^{241}$A) or TAK1 ($N^{245}$A) had lower $Il6$ production when stimulated with poly(I:C) (Fig. 5p), which

indicates association with PIP2 is crucial for TAK1 activation in the TLR3 signaling pathway.

**PLCβ2-deficient mice are more susceptible to CVA16 infection.** The massive induction of proinflammatory cytokines, such as IL-6 could be responsible for severity of pathogenesis in HFMD[26,31,34]. As $Plcb2^{-/-}$ mice had significantly higher mRNA levels of $Tnf$ (Fig. 2a), $Il6$ (Fig. 2b) and $Il12$ (Fig. 2c) and little changed viral loads (Supplementary Fig. 6a) compared with wild-type mice after CVA16 infection, we next monitored the survival and histopathological pathogenesis of mice challenged with CVA16. CVA16-infected $Plcb2^{-/-}$ mice died more rapidly than wild-type mice (Fig. 6a) and had a higher clinical

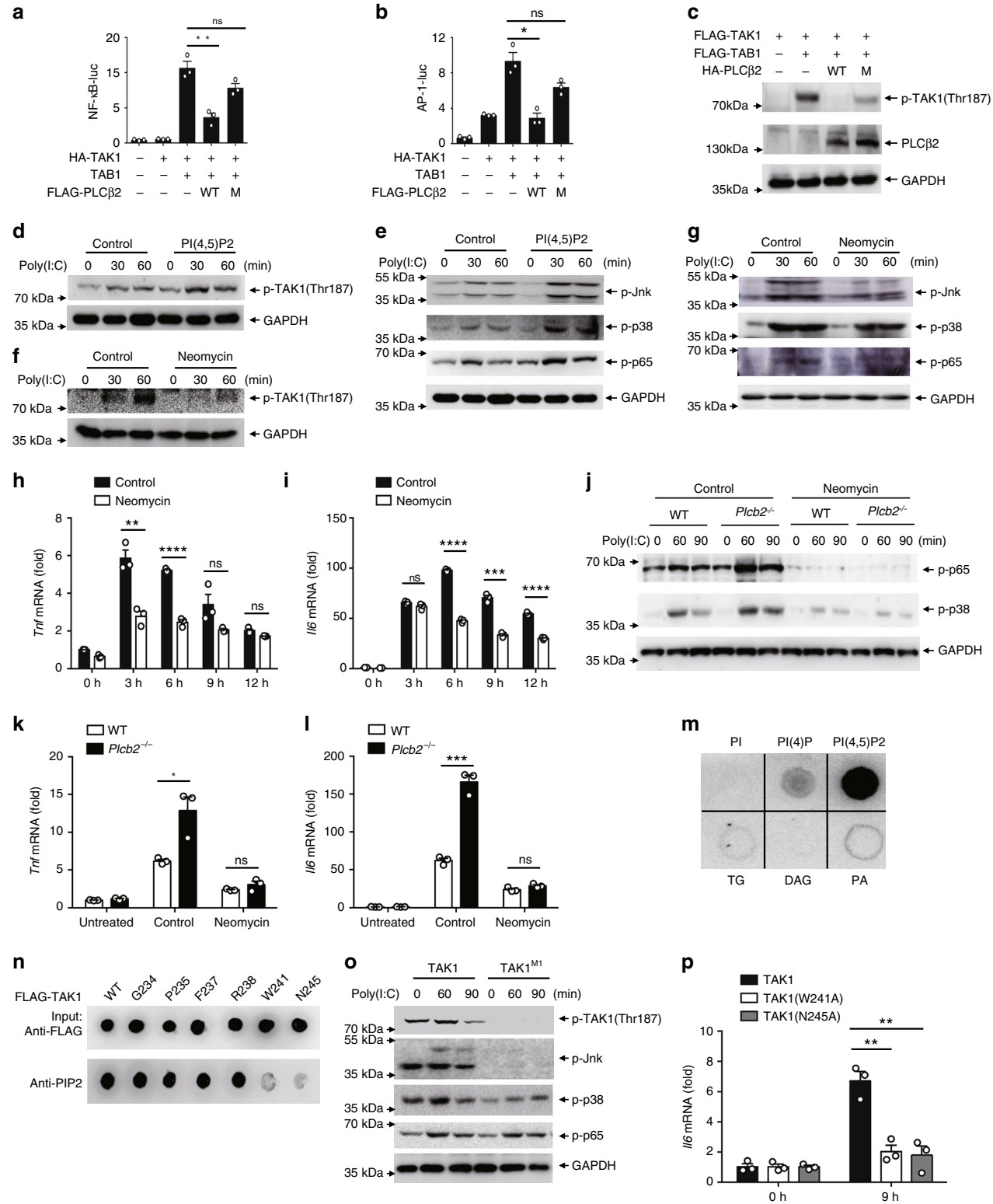

score compared with wild-type mice (Fig. 6b). Histological examination of the lung or skeletal muscle tissue isolated from mice challenged with CVA16 showed that *Plcb2*−/− mice exhibited increased histopathological pathogenesis compared with wild-type mice (Fig. 6c, d). The fact that CVA16 infection induced more cytokines and death in PLCβ2-deficient mice suggests that PLCβ2 may function as an important modulator of pathogenesis of CVA16 infection in vivo.

**M-3M3FBS alleviates the pathogenesis of CVA16 infection.** The agonist 2,4,6-trimethyl-N-(meta-3-trifluoromethyl-phenyl)-benzene sulfonamide (m-3M3FBS) has been reported to directly stimulate PLC activity[35]. Since we have identified PLCβ2 as a negative regulator of proinflammatory cytokines during viral infection, we next evaluated the effect of m-3M3FBS on virus-induced immune responses. Treatment with m-3M3FBS mark-edly suppressed TAK1 phosphorylation (Fig. 7a), as well as

**Fig. 5** PLCβ2 inhibits TAK1 activation via PIP2. **a, b** Reporter assay of NF-κB (**b**) and AP-1 (**c**) activation in HEK293T cells expressing TAK1 and TAB1 in the presence or absence of PLCβ2 (WT) or phospholipase-inactive mutant (M). **c** IB of cell lysates from HEK293 cells expressing TAK1 and TAB1 in the presence or absence of PLCβ2 (WT) or phospholipase-inactive mutant (M). **d–g** IB of cell lysates from macrophages treated with PIP2 in a lipid carrier for 10 min or 10 mM neomycin for 2 h before stimulating with poly(I:C) for the indicated times using anti-phospho-antibodies. **h, i** Q-PCR analysis of relative *Tnf* (**h**) and *Il6* (**i**) mRNA levels in macrophages pretreated with neomycin before stimulating with poly(I:C) for the indicated time. **j** IB of cell lysates from wild-type or *Plcb2*$^{-/-}$ macrophages treated with 10 mM neomycin for 2 h before stimulating with poly(I:C) for the indicated times using phospho-antibodies. **k, l** Q-PCR analysis of relative *Tnf* (**k**) and *Il6* (**l**) mRNA levels in wild-type or *Plcb2*$^{-/-}$ macrophages pretreated with 10 mM neomycin for 2 h before stimulating with poly(I:C) for 6 h. **m** Phospholipid-binding assays with purified FLAG-TAK1 protein from HEK293 cells. PI phosphatidyl- inositol, TG triglyceride, DAG diacylglycerol, PA phosphatidic acid. **n** Dotting assay of PI(4,5)P2 with purified TAK1 and mutants. **o, p** *TAK1*$^{-/-}$ A549 cells complemented with wild-type TAK1 or its W241A or N245A mutants and stimulated with poly(I:C) for the indicated times before assay via immunoblotting with anti-phospho-antibodies (**o**) or Q-PCR analysis of relative *Il6* mRNA (**p**). ns not significant ($p > 0.05$); *$p < 0.05$; **$p < 0.01$; ***$p < 0.001$ and ****$p < 0.001$ by unpaired *t*-test (**a, b, h, i, k, l** and **p**). Data are representative of three experiments with at least three independent biological replicates (mean and s.e.m. of $n = 3$ cultures in **d–l**)

activation of the MAPKs and NF-κB pathways (Fig. 7b) in poly(I:C)-stimulated macrophages. Similarly, m-3M3FBS markedly reduced the expression of proinflammatory cytokines including *Tnf* (Fig. 7c), *Il6* (Fig. 7d) and *Il12* (Fig. 7e) in poly(I:C)-stimulated macrophages. Furthermore, treatment with m-3M3FBS dramatically reduced the clinical score (Fig. 7g) and prolonged the survival of CVA16-infected mice (Fig. 7f). Histopathology was also reduced in both lung and skeletal muscle tissue (Fig. 7h, i). Additionally, expression of wild-type and enzyme mutant PLCβ2 abolished m-3M3FBS-mediated TAK1 phosphorylation inhibition, which suggests that m-3M3FBS modulates TAK1 activation via PLCβ2 (Supplementary Fig. 6b). These results suggest that m-3M3FBS and PLCβ2 could be a potential effective drug candidate and target, respectively, for the control of HFMD and other inflammatory diseases.

## Discussion

Antiviral immunity is regulated at multiple steps in the signaling cascade. Despite considerable evidence that PLCβ2 is a crucial enzyme required for effective signal transduction and leukocyte activation[20], relatively little is known about how PLCβ2 might regulate antiviral innate inflammatory responses. In this study, we demonstrated that PLCβ2 inhibits poly(I:C)-induced and virus-induced production of proinflammatory cytokines in vitro and in vivo. Mechanistically, PLCβ2 interacts with TAK1 and may inhibit TAK1 activation by hydrolyzing its substrate PIP2, which is found to bind with TAK1 at W$^{241}$ and N$^{245}$ and promotes the activation of TAK1. Thus, PLCβ2 negatively regulate virus-induced pro-inflammatory responses by inhibiting phosphoinositide-mediated activation of TAK1.

PLCβ isoforms are activated by heterotrimeric GTP-binding proteins linked to various cell surface receptors[20]. In innate immune cells, it has been reported that LPS suppresses the expression of PLCβ2 in macrophages through a MyD88-dependent pathway[24]. Consistent with this finding, we also observed the downregulation of PLCβ2 by LPS stimulation in our experiments (Supplementary Fig. 1d). However, our data demonstrated that poly(I:C) stimulation-induced PLCβ2 expression, apparently through the TLR3-p38-signaling pathway. Infection with CVA16 also induced the expression of PLCβ2. Importantly, the PLCβ2 protein level was found to be significantly higher in patients with HFMDs. These results indicate that the regulation of PLCβ2 expression is complicated and varies by type of infection or stimulation and requires further investigation.

TAK1 has been reported to be an indispensable kinase for TLR3-mediated activation of NF-κB and AP-1[36]. Other reports indicate that TAK1 is essential for the activation of NF-κB, p38, and JNK by HSV, Sendai virus, and RSV[37–39]. Our data indicate that overexpression of PLCβ2 inhibits TAK1 phosphorylation, as well as TAK1-mediated activation of NF-κB and AP-1.

Furthermore, we found that PLCβ2-deficient macrophages display enhanced TAK1 activation after poly(I:C) stimulation, which is accompanied by increased activation of NF-κB and phosphorylation of MAP kinases. By contrast, treatment with m-3M3FBS, a PLC activator, suppressed poly(I:C)-induced TAK1 phosphorylation and its downstream signaling events. Thus, it is possible that TAK1 is involved in the activation of NF-κB and AP-1 in macrophage cells stimulated with dsRNA, a process in which PLCβ2 plays a role as an essential negative regulator.

PLCβ2 is known to hydrolyze PIP2 into IP3 and DAG, which mediates the release of Ca$^{2+}$ from intracellular stores, or activates PKC to regulate downstream signaling events[18–20]. PIP2 regulate multiple functions in cellular events, and its concentration is controlled by polyphosphoinositide kinase and lipid-metabolizing enzymes in response to many extracellular stimuli[18]. Upon LPS stimulation, PIP2 is generated by PI4-phosphate5-kinase (PIP5K), and mediate the translocation of Toll/IL-1 receptor domain-containing adaptor protein (TIRAP) to the plasma membrane, which is required for the activation of TLR4-signaling pathway[40–42]. However, TIRAP is probably not involved in the TLR3-mediated signaling because TLR3 does not interact with TIRAP after poly(I:C) stimulation. TLR3-mediated signaling relies on the adaptor TRIF that is responsible for recruitment of the TRAF6–TAB2–TAK1 complex[43–45]. Here we found that PIP2 promotes the interaction of TAK1 with TAB1, and activation of TAK1 in the poly(I:C)-induced TLR3-signaling pathway. Besides, PLCβ2 is found to interact with and inhibit TAK1 via its phosphodiesterase activity. Thus, these results suggested that PLCβ2 may disrupt the interaction of TAK1 with TAB1 through hydrolyzing PIP2, thus inhibiting the activation of TAK1 in TLR3 signal. Moreover, our data indicate that PIP2 directly binds with TAK1 at W$^{241}$ and N$^{245}$, and impairing of PIP2's binding affinity or mutation of PIP2-binding sites on TAK1 abolished its activation and the subsequent production of pro-inflammatory cytokines. TAK1 is a highly conserved MAPK kinase kinase that is subjected to multiple levels of modification and plays a pivotal regulator of innate immune responses[11–17]. To the best of our knowledge, our finding is the first time to demonstrate a phosphoinositide-mediated activation of MAP kinase kinase. However, whether PIP2-dependent activation of TAK1 requires the recruitment of TAK1 to the plasma membrane awaits further investigation.

HFMDs caused by enteroviruses, such as CVA16 and EV71, result in millions of infections and hundreds of deaths each year and are now recognized as important emerging infectious diseases[2]. High levels of IL-6 are reported to contribute to the pathogenesis of HFMDs[28–31]. Although higher amounts of proinflammatory cytokines induced by pathogens are thought to recruit and activate the acute-phase response and T and B cell for pathogen clearance, accumulating evidence indicate that high levels of proinflammatory cytokines can overwhelm immune

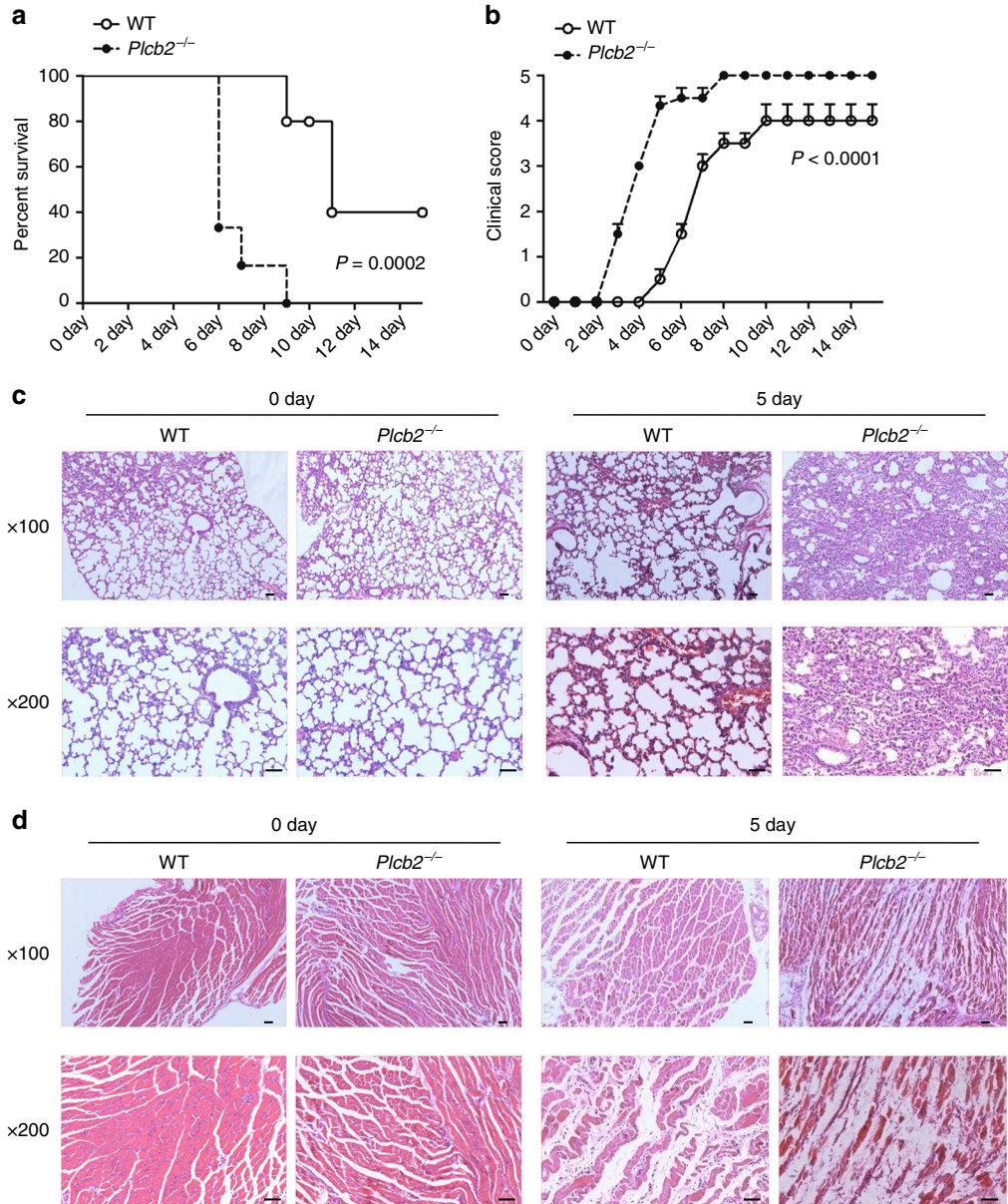

**Fig. 6** PLCβ2-deficient mice are more susceptible to CVA16 infection. Fourteen-day-old wild-type or *Plcb2*−/− mice were infected intraperitoneally with 50 μl of CVA16 (1.5 × 10⁴ PFU/mouse) for 0 or 5 days. Survival (**a**), clinical score (**b**), or histopathology of lung (**c**), or skeletal muscle (**d**) tissue of wild-type or *Plcb2*−/− mice (n = 6) treated as indicated. Scale bar, 50 μm. Data are representative of three experiments with at least three independent biological replicates (mean and s.e.m. of n = 6 mice per group). Gehan–Breslow–Wilcoxon test (**a**) and two-way analysis of variance plus Bonferroni's posttest (**b**) were used to calculate p value

defenses and cause severe inflammatory diseases[1,3–5,28,46]. In this study, our data indicates that PLCβ2 is highly induced in clinical samples from HFMDs patients or CVA16-infected macrophages. PLCβ2-deficient mice infected with CVA16 have higher levels of proinflammatory cytokines and are more likely to succumb to virus infection. These results suggest that PLCβ2 may function as a negative feedback inhibitor to attenuate the CVA16 virus-induced production of pro-inflammatory cytokines in vivo. Furthermore, treatment with the PLC activator, m-3M3FBS, reduced the histopathology of CVA16-infected mice in both lung and skeletal muscle tissue and prolonged the survival of CVA16-infected mice. Thus, targeting PLCβ2 may be beneficial in the treatment of viral diseases such as HFMDs.

In summary, we have shown that PLCβ2 is specifically upregulated by dsRNA and virus infection. PLCβ2 interacts with the TAK1–TAB1 complex and suppresses the phosphorylation and activation of TAK1, which may be mediated by hydrolyzing PIP2 (Fig. 7j). Our model explains why CVA16 infection induced more cytokines and caused more death in PLCβ2-deficient mice. These findings provide insights into the negative regulation of antiviral innate immune responses. To our knowledge, this is the first report of a functional role of PIP2 in the TLR3 signal pathway. In addition, our data provide a mechanistic explanation for the pathogenesis of viral infection and highlight potential therapeutic targets for virus-associated inflammatory diseases.

## Methods

**Cell culture and reagents**. HEK293T (ATCC CRL-3216) cells were obtained from the American Type Culture Collection (ATCC) and maintained in Dulbecco's modified Eagle's medium (DMEM, Hyclone) supplemented with 10% (v/v) heat-inactivated fetal bovine serum (FBS, Gibco) and 100 U/ml penicillin and streptomycin. Peritoneal macrophages were cultured in RPMI-1640 medium supplemented with 10% (v/v) FBS and 100 U/ml penicillin and streptomycin. Vero cells

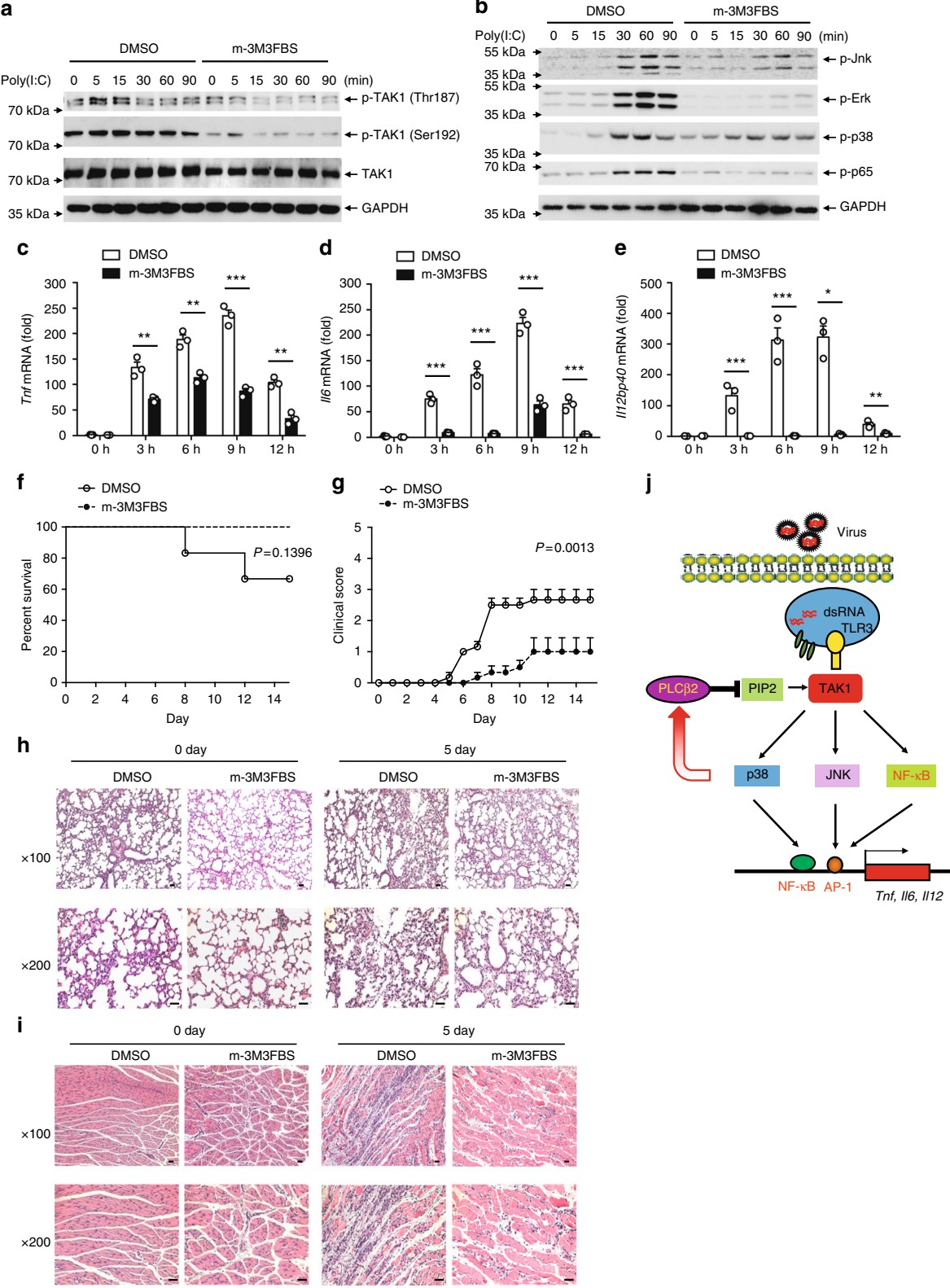

**Fig. 7** PLCβ2 activator alleviates the pathogenesis of CVA16 infection. **a**, **b** Immunoblot analysis of lysates from macrophages pretreated with DMSO or m-3M3FBS, and then stimulated with poly(I:C) for the indicated times using different phospho-antibody. **c–e** Q-PCR analysis of relative *Tnf* (**c**), *Il6* (**d**) and *Il12p40* (**e**) mRNA in macrophages pretreated with DMSO or m-3M3FBS, then stimulated with poly(I:C) for the indicated times. Fourteen-day-old wild-type mice were pretreated with DMSO or m-3M3FBS and subsequently infected with CVA16 for 0 or 5 days. **f–i** Survival rate (**f**), clinical score (**g**) and histopathology in lung (**h**) or skeletal muscle (**i**) tissue of mice treated as above at either day 0 (not infected) or day 5. Scale bar, 50 μm. **j** Diagram. *$p < 0.05$; **$p < 0.01$ and ***$p < 0.001$ by unpaired *t*-test (**c–e**). Gehan–Breslow–Wilcoxon test (**f**) and two-way analysis of variance plus Bonferroni's posttest (**g**) were used to calculate *p* value. Data are representative of three experiments with at least three independent biological replicates (mean and s.e.m. of *n* = 3 cultures in **a–e** or *n* = 6 mice per group in **f–h**)

(Africa green monkey kidney cells) were cultured in a VP-SFM medium (Gibco) supplemented with 4 mM L-glutamine (Gibco). Cells were maintained in a 37 °C incubator equilibrated with 5% CO₂. All the cells were routinely tested for contamination by mycoplasma. A clinically isolated CVA16 strain was propagated in Vero cells using the microcarrier cell culture bioprocess, which has been previously reported. Virus stocks were stored at −80 °C. Viral titers were tested using a standard plaque-forming assay. The following antibodies were used: rabbit anti-K63-linkage specific polyubiquitin (5621), rabbit anti-phospho-TAK1 (Thr187) (4536), rabbit anti-phospho-p65 (3033), rabbit anti-phospho-Erk1/2 (9101), rabbit anti-phospho-p38 (9215), and rabbit anti-phospho-Jnk (9251), all from Cell Signaling Technology; rabbit anti-PLCβ2 (sc-206), rabbit anti-phospho-TAK1 (Ser192) (sc-130219), rabbit anti-TAK1 (sc-7162), and monoclonal mouse anti-TAK1 (sc-7967) all from Santa Cruz Biotechnology; rabbit anti-TAB1 (ab27983, from Abcam); rabbit anti-EEA1 (PA1-063A, from Invitrogen) monoclonal mouse anti-FLAG M2-affinity gel (A2220), monoclonal mouse anti-HA (H9658), rabbit anti-HA (H6908), rabbit anti-GAPDH (SAB2701826), and rabbit anti-FLAG (F7425), all from Sigma; and Protein G Sepharose™ 4 Fast Flow (GE Healthcare). The following compounds were used: m-3M3FBS (T5699), PDTC (NF-κB inhibitor, P8765), PD98059 (MEK inhibitor, P215), SP600125 (JNK inhibitor, S5567), and SB203580 (P38 inhibitor, S8307), all from Sigma. Resiquimod r-848 (tlr7 ligand) (S8133, from selleck).

**Plasmids and plasmid construction**. The cDNAs encoding PLCβ2 and its truncated forms, as well as the cDNAs encoding TAK1 and its deletion mutants were inserted into the FLAG-pcDNA3 vector. PLCβ2 and TAK1 cDNAs were inserted into the HA-pcDNA3 expression vector. The cDNAs encoding TAB1 were inserted into the pcDNA3 vector, and cDNAs encoding PLCβ2 and TAK1 were inserted into the pET28a and p-GEX-4T-1 vectors, respectively.

**Mouse strains**. Homozygous *Plcb2* knockout and *Tlr3* knockout mice with C57BL/6 bacground were bred in specific pathogen-free conditions at the Shanghai Laboratory Animal Center. Fourteen-day-old or 6-week-old *Plcb2* knockout mice and wild-type control mice were used in the experiments. All animal studies were approved by the Institutional Animal Care and Use Committee of Tongji University.

**RT-PCR analysis**. Initially, cells were incubated for 12 h without serum. After preincubating with DMSO or m-3M3FBS (10 mM) for 30 min, the cells were stimulated with poly(I:C)(10 mg/ml) for the indicated time periods. Total RNA was extracted with 1 ml of TRIzol reagent according to the manufacturer's instructions (Invitrogen). Next, 1 μg of total RNA was reverse transcribed using the ReverTra Ace® qPCR RT Kit (Toyobo, FSQ-101) according to the manufacturer's instructions. A LightCycler (Roche, LC480) and a SYBR RT-PCR kit (Toyobo, QPK-212) were used for quantitative real-time RT-PCR analysis. The expression values were normalized to those obtained with the control gene Gapdh (encoding GAPDH). The primers used were as follows: CVA16 forward and reverse (5′-GAACCATC ACTCCACACAGGAG-3′ and 5′-GTACCTGTGGTGGGCATTG-3′, respectively), GAPDH forward and reverse (5′-CCCACTAACATCAAATGGGG-3′ and 5′-CC TTCCACAATGCCAAAGTT-3′, respectively), IL-6 forward and reverse (5′-TC CAGTTGCCTTCTTGGGAC-3′ and 5′-GTGTAATTA AGCCTCCGACTTG-3′, respectively), TNF-α forward and reverse (5′-TTCTGTCTACTGAACTTCGGG GTGATCGGTCC-3′ and 5′-GTATGAGATAGCAAATCGGCTGACGGTGTGG G-3′, respectively), IL-12p40 forward and reverse (5′-GAGCACTCCCCATTCCT ACT-3′ and 5′- CCCTCCTCCTGTCTCCTTCAT-3′, respectively), PLCβ1 forward and reverse (5′-GTGCACAGAGGATGTGCTGA-3′ and 5′-CCAAGTGTCCGA TGTTCCCA-3′, respectively), PLCβ2 forward and reverse (5′-GCTTCCTCTCCT GTTCACCC-3′ and 5′-CCTTCACGTTAGGGGGCAAT-3′, respectively), PLCβ3 forward and reverse (5′-GGAGCGTGTGGAGAGAGCAG-3′ and 5′-AGCACTT CGTTGAGTCTCGG-3′, respectively), PLCβ4 forward and reverse (5′-ACCCGCT GGCTCATTACTTC-3′ and 5′-TGATCAGCACTTTCTATTTCCTCT-3′, respectively).

**Small interfering RNA-mediated gene interference**. Three small interfering RNAs (siRNAs) sequence were designed to silence one gene, and peritoneal macrophages were transfected three siRNA mixture with siRNA-Mate transfection reagent (G04003, from GenePharma) according to the manufacturer's instructions. The targeting sequences were: mouse PLCβ1: siRNA1, 5′-CCUCAAGAAGGGCA CCAAATT-3′; siRNA2, 5′-CCAAGUGUUGAUUGAGAAATT-3′; siRNA3, 5′-GC UUGUAGUCUUCCAUCUUTT-3′. mouse PLCβ2: siRNA1, 5′-CCAGAGGACUU CCCAGAAUTT-3′; siRNA2, 5′-CCACCAAGACAUGACACAATT-3′; siRNA3, 5′-GCAUGGACUCUUCCAACUAUTT-3′. mouse PLCβ3: siRNA1, 5′-CCACAUU CUUGAACUUCAUTT′; siRNA2, 5′-GCUUCUGGAGAUGUCAUCCUUTT-3′; siRNA3, 5′-GCCACAAGGCUAUGGUGAATT-3′. mouse PLCβ4: siRNA1, 5′-G CACGGAUCUGGUGAAUAUTT′; siRNA2, 5′-GCGACAAAUGAGCCGCAUU TT-3′; siRNA3, 5′-CCUGACAACGGAUCACAAATT-3′.

**Dual-luciferase reporter assay**. The mammalian HEK293T cell line was used for the dual-luciferase reporter assay. The cells were cultured in DMEM (high glucose; Gibco, Life Technologies, Grand Island, NY, USA) with 10% FBS. Before

transfection, the cells were plated in a 96-well tissue culture plate. After 24 h, two reporter plasmids (total of 10 ng per well) and the indicated plasmids (100 ng per well) were co-transfected using Lipofectamine™ 2000 Transfection Reagent (Invitrogen, Life Technologies, Grand Island, NY, USA). Approximately 48 h after transfection, Renilla and firefly luciferase activities were quantified using the Luciferase Assay Reagent II (LAR II) (Promega, Madison, WI, USA) on a Modulus™ single-tube multimode reader (Turner Biosystems, Sunnyvale, CA, USA) according to the manufacturer's instructions. The transfections were performed in triplicate. A paired t-test was used to determine significant differences.

**Western blot and immunoprecipitation**. For immunoprecipitation, HEK293T cells were transiently transfected using the calcium phosphate–DNA co-precipitation method. After 48 h, the cells were washed with PBS and subsequently lysed in lysis buffer (20 mM Tris (pH 7.5), 150 mM NaCl, 1% Triton X-100, sodium pyrophosphate, β-glycerophosphate, EDTA, Na3VO4, leupeptin) supplemented with 1% protease inhibitor cocktail (Sigma, P8340), 1 mM NaF, and 1 mM Na3VO449. After 30 min on ice, the lysates were centrifuged at 13,523g for 15 min at 4 °C to remove debris. The cell lysates were incubated with anti-Flag M2-affinity gel overnight at 4 °C. For endogenous immunoprecipitation, peritoneal primary macrophages were stimulated with poly(I:C) for the indicated time periods. The cells were subsequently lysed, and the lysates were incubated with anit-PLCβ2 antibody and Protein G Sepharose™ 4 Fast Flow overnight at 4 °C. The sepharose samples were centrifuged, washed three times with cell lysis buffer, and boiled with SDS loading buffer for 8 min. The original western blots from the main figures are shown in Supplementary Figure 7.

**GST fusion proteins and precipitation assay**. The TAK1 and PLCβ2 proteins were amplified by PCR and subcloned into the pGEX-4T1 (Amersham) and pET28a (Novagen) vectors, respectively. The His and GST fusion proteins were expressed in BL-21(DE3) bacteria (Invitrogen) according to the manufacturer's instructions. The RAW264.7 cells were lysed as described above. The lysates or purified His-PLCβ2 protein were incubated for 4 h at 4 °C with equal amounts of the appropriate fusion protein coupled to glutathione beads. The beads were isolated by centrifugation, washed, boiled with SDS loading buffer for 8 min, and later analyzed by immunoblot.

**Cell staining and confocal microscopy**. Peritoneal macrophage stimulated with poly(I:C) for 4 h, and the cells were fixed with 4% formaldehyde for 30 min at 25 °C. After permeabilized for 10 min with 0.1% Triton X-100 in PBS, the samples were blocked with 3% BSA in PBS for 30 min at 25 °C. The cells were incubated with the indicated antibodies overnight at 4 °C. Following PBS wash for three times, Invitrogen Alexa Fluor 488 or 555 conjugated secondary antibodies were used to stain the cells. The cells were examined using a Leica confocal microscope equipped with analytical software[47].

**Animal experiments**. Specific-pathogen-free, 14-day-old wild-type or *Plcb2* knockout mice were intraperitoneally (i.p.) injected with 50 μl of CVA16 (1.5 × 10⁴ PFU/mouse) using a needle. The mice were observed once daily for clinical signs and mortality for 14 days. DMSO or m-3M3FBS (5 mg/kg) was injected three times into CVA16-infected mice (1 day before, 2 days after, and 5 days after infection). Clinical disease was scored as follows: 0, healthy; 1, ruffled fur and hunchbacked appearance; 2, wasting; 3, limb weakness; 4, limb paralysis; 5, moribund and death. At 5 days after infection lung and skeletal muscle tissue samples were collected from wild-type or PLCβ2 knockout mice. Tissues were fixed in 10% neutral-buffered formalin (Sigma-Aldrich) and paraffin embedded, sectioned, and stained with hematoxylin and eosin (H&E). Bright field microscopy images were obtained at ×100 or ×200 magnification. All the mice were age-matched and sex-matched in each experiment. No animals were excluded from the study, and sample size was based on empirical data from pilot experiments. No additional randomization or blinding was used to allocate experimental groups.

**PI(4,5)P2 delivery**. For delivery of PI(4,5)P2 into macrophages, PI(4,5)P2 and Carrier 3 (Echelon Biosciences Inc.) were preincubated at a 1:1 molar ratio (100 mM final concentration each) for 10 min at room temperature and then added to macrophage cells or HEK293T cells at a final concentration of 10 mM. After incubation for 10 min at 37 °C, the cells were stimulated with poly(I:C) for the indicated time.

**Ethics statement**. This work received approval from the Clinical Ethics Committee of Xinhua Hospital, which is affiliated with Shanghai Jiaotong University School of Medicine, in accordance with governmental guidelines and institutional policies. Written informed consent was obtained from the next of kin, caretakers, or guardians on behalf of the minors/children participants involved in our study. The ethics committee specifically approved this consent procedure.

**Human samples**. Whole blood samples were obtained from children under 10 years old. Children were clinically diagnosed with HFMD according to the Ministry of Health diagnostic criteria of HFMD (2008), China (http://www.moh.gov.cn/

mohbgt/s9503/200812/38494.shtml). Additionally, age-matched healthy children with no history of HFMD were included as controls. For western blotting and mass spectrum assay, whole blood samples were treated with red blood cell lysis buffer (eBioscience) according to the manufacturer's instructions; subsequently, the remaining cells were lysed in lysis buffer (as indicated). In mass spectrum assay, samples were divided into two groups, control and HFMD, and each group was the mixture of different samples.

**Protein extraction and iTRAQ labeling**. The obtained white blood cells from control and HFMD samples were lysed in buffer containing 7 M urea, 2 M thiourea, 1 mM phenylmethanesulfonyl fluoride, and protease inhibitor cocktails, respectively. The cell lysates were centrifuged at 12,000×g for 40 min at 4 °C to collect the supernatant. Protein concentration was determined using Bradford assay. Then, 100 μg proteins of each sample were reduced in 50 mM tris-(2-carboxyethyl) phosphine (TCEP) at 56 °C for 1 h and alkylated in 200 mM methyl-methanethiosulfonate (MMTS) at room temperature for 1 h. The reduced and alkylated protein mixtures were precipitated by adding 5× volume of chilled acetone at −20 °C overnight. After centrifugation at 4 °C, 30,000 × g for 30 min, the pellet was dissolved in 0.5 M triethylammonium bicarbonate (TEAB) solution and sonicated in ice. Samples were then proteolyzed with 1 μg trypsin: 20 μg proteins at 37 °C for 16 h. Each digested sample was labeled with iTRAQ reagent at room temperature for 2 h according to the manufacturer's instructions (Applied Biosystems, Framingham, MA, USA). As a consequence, 1 set of iTRAQ 2-Plex (control with 115 tags, HFMD with 116 tags) was constructed. The labeled peptides of two groups were mixed, respectively, at 1:1, dried on a rotation vacuum concentrator (Christ, Germany) for further analysis.

**2DLC–MS/MS analysis of iTRAQ-labeled peptides**. The labeled peptides were first fractionated using high pH reversed phase liquid chromatography (RPLC) on a UPLC system (Waters, USA). The peptides were re-suspended with loading buffer (5 mM ammonium formate containing 2% ACN, pH 10) and separated on a C18 column (Waters, 3.5 μm, 2.1 × 250 mm Xbridge BEH300) with a flowrate of 300 μl/min. The gradient elution was performed by 0–25% B (5 mM ammonium formate containing 98% ACN, pH 10, 5–35 min) and 25–45% B (35–48 min). A total of 20 fractions were collected and then mixed to 12 fractions for each iTRAQ group set. All the fractions were dried on a rotation vacuum concentrator (Christ, Germany). The 2-Plex-labeled samples were analyzed by 1D Plus nano LC system (Eksigent, USA) coupled with Triple TOF 5600 system (Applied Biosystems, USA) with a 15-cm-long column (75 μm i.d., packed with 5 μm i.d. 100 Å pore size, C18 packing materials, Thermo Scientific, USA). The following gradient conditions were used with phase B (100% ACN with 0.1% formic acid): 5% B (0–0.1 min), 5–30% B (0.1–70 min), 30–80% B (70–71 min), 80% B (71–75 min), 80–5% B (75–75.1 min), 5% B (75.1–90 min), and total flow rate was maintained at 300 nl/min. Electrospray voltage of 2.3 kV versus the inlet of the mass spectrometer was used. Triple TOF 5600 mass spectrometer was operated in information-dependent mode to switch automatically between MS and MS/MS acquisition. MS spectra were acquired across the mass range of 350–1200 m/z in high-resolution mode using 250 ms accumulation time per spectrum. Tandem mass spectral scanned from 100 to 1200 m/z in high-sensitivity mode with rolling collision energy. The 20 most intense precursors were selected for fragmentation per cycle with dynamic exclusion time of 25 s.

**Protein identification and quantitation**. Protein identification and quantification for iTRAQ experiments was carried out using ProteinPilot 4.5 software (Applied Biosystems, MDS-Sciex). Database search was performed against a target-decoy database constructed based on a SwissProt human (22706 sequence entries). Searching parameters were set as following: Quantitation: iTRAQ-4plex; Enzyme: Trypsin; Missed cleavage: 1; Variable modification: Oxidation (Met); Fixed modification: Methyl methanethiosulfonate (Cys). MS1 tolerance: 20 ppm; MS/MS tolerance: 0.1 Da. False discovery rate lower than 1% was used to control protein level identification based on the target-decoy strategy. Proteins with at least one unique peptide with confidence higher than 95% were used for quantitation. Proteins with a fold change larger than 1.2 or less than 0.8 were selected as differently expressed proteins.

**Statistical analysis**. Statistical significance between the groups was determined using a two-tailed Student's t-test and a two-way analysis of variance (ANOVA). Differences were considered to be significant when $p < 0.05$. For the mouse survival study, Kaplan–Meier survival curves were generated and analyzed for statistical significance by the Gehan–Breslow–Wilcoxon test with GraphPad Prism 6.0. Clinical scores between two groups were determined by two-way ANOVA plus Bonferroni's posttest.

**Reporting summary**. Further information on experimental design is available in the Nature Research Reporting Summary linked to this article.

## Data availability

The authors declare that data supporting the findings of this study are available within the manuscript and Supplementary Information.

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

## Acknowledgements

We thank J. Yang (Baylor College of Medicine) for providing HA-Ub(K63) constructs. We thank members of the B. Ge laboratory for helpful discussions and technical assistance. This work was supported by grants from Chinese National Program on Key Basic Research Project (2017YFA0505900, 973 Programs 2012CB578100 and 2011CB505000) National Natural Science Foundation of China (31730025, 31670901, 91542111, 81330069, and 81800004), the National Key Research and Development Program of China (program 2016YFC1305103), the Program for Professor of Special Appointment (Eastern Scholar) at Shanghai Institutions of Higher Learning (program TP2016007), Shanghai Pujiang Program (program 16PJ1401400), the Outstanding Youth Training Program of Shanghai Municipal Commission of Health and Family Planning (program 2017YQ012) and the Fundamental Research Funds for the Central Universities (program 22120180024).

## Author contributions

L.W., Y.Z., D.Y. and B.G. designed this study. L.W., Y.Z., Z.C. and D.Y. performed experiments, assisted by L.S., J.W., H.L, F.L. and F.W. Mice infection experiments was performed by C.Y., J.Y. and Q.L. Q.Z., A.X., L.S. and J.S. collected clinical samples. D.W. provided PLCβ2 knockout mice. C.F. and H.L. proformed MS analysis. L.W., Y.Z., D.Y. and B.G. analyzed the data and wrote the manuscript. All authors discussed the results and commented on the manuscript.

## Additional information

**Competing interests:** The authors declare no competing interests.

