## [Peer Review File · Nature Communications]

Reviewers' comments:

Reviewer #1 (Remarks to the Author):

This manuscript describes an interesting study of the role of PLC β 2 in inflammatory responses to RNA viruses (enteroviruses) via negative regulation of TAK1 kinase. First of all, PLC β 2 expression was increased in vivo in hand-foot-mouth disease (HFMD) patients and in vitro in CVA16-infected macrophages via TLR3 and p38. Second, mRNA expression of proinflammatory cytokines (TNF, IL-6 and IL-12) were enhanced in the lungs of CVA16-infected Plcb2 $^{-/-}$ mice, and TNF and IL-6 mRNAs and phosphorylation of MAPKs and NF- κ B were increased in CVA16-infected macrophages. Third, physical interactions of PLC β 2 with TAK1 were shown. Fourth, PLC β 2 inhibits TAK1 phosphorylation and activity. An inverse correlation of PLC β 2 expression with TAK1 phosphorylation was found in HFMD patients. Fifth, PLC β 2-mediated inhibition of TAK1 activity is via reduction of PIP2, as PIP2 activates TAK1. Sixth and finally, the enhanced susceptibility to CVA16 virus was found in Plcb2 $^{-/-}$ mice, and on the other hand, PLC activator m-3M3FBS protected mice from CVA16 mice. Overall, this is a novel and well-executed study. However, I have issues that preclude my endorsement of the study right away: Nothing has been done on potential redundant functions of other PLC β isoforms. This is particularly important because PIP2 reduction seems a critical factor that regulates TAK1 activity. Along the same line, nothing has been done to address whether direct interaction of PLC β 2 with TAK1 plays any role in the regulation of TAK1 activity.

Many minor issues:

There is no description on mass spec and no explanation on Fig. 1a.

Fig. 3f. There is no baseline control for IP (with non-specific antibody).

Fig. 5. PLC β 2 mutant was not described in sufficient detail. Downstream phenomena used in Fig. 5f, g, o are arbitrarily selected.

Fig. 6a,b. Stats are missing.

Fig. 7f,g. Stats are missing.

Line 122. "stimulus" should read "stimuli".

Line 172. "stimulus-dependent". This is not accurate. It should be "constitutive, but affected by poly(I:C) stimulation" or similar.

Line 178. "greatly inhibited" sounds exaggerating.

Lines 212-213. "wild-type PLC β 2, but not....." This sounds like "all or none" phenomenon.

Line 217. "much higher" sounds exaggerating.

Lines 229, 236. "a dot blot assay" seems better.

Line 234. Explain "MODA".

Line 255. What muscle was analyzed?

Line 331. "multiple levels"

Line 342. "T and B cells"

Reviewer #2 (Remarks to the Author):

In this study, Wang et al find that RNA virus infection could induce the expression of PLC β 2, which in turns interacts with TAK1 and suppress its activation. The authors then revealed that PIP2, substrate of PLC β 2, is responsible for the activation of TAK1 and subsequent production of pro-inflammatory cytokines in intro and in vivo. This study revealed a novel mechanism of PIP2 in regulating inflammatory response caused by virus infection, which have broad interests to the field of antiviral innate immune response. The reviewer has the followor comments:

1. The authors have shown the interaction between PIP2 and TAK1 was essential for TAK1 activation. Moreover, PLC β 2 could disrupt TAK1-TAB1 interaction. is there any link between them?
2. The details about Mass Spectrum analysis should be provided. The experiment was performed with a mixture of different samples or not? The downregulated proteins were not needed in Fig. 1a. The meaning of protein level of "control" and p value in Fig.1b should be clearly stated.
3. Line 127, the authors should explain why they focused on Tlr3 but not other RNA sensors like Tlr7, RIG-I or MDA5. The conclusion about this experiment should be accurate.
4. The authors showed that Mutant of PLCb2 could not suppress TAK1 activation, how about the binding activity between TAK1 and the mutant? Does the mutant keep the ability to inhibit TAK1-TAB1 interaction?
- 5.Line 239, "Tak1-/- A549 cells" should be "TAK1-/- A549 cells". The expression of TAK1 mutants should be illustrated.
6. Line 232, the details of MODA should be provided. Since TAK1 mainly located in cytosol, the authors should explain why TAK1 could bind PIP2 and where the binding occurs.
7. Line 421, the genetic background of mice should be mentioned.
8. Fig. 6 and 7, viral loads data should be provided to confirm whether the pathogenesis were caused by cytokine storm or virus replication.
9. Fig. s6 should be combined into main figure.

Reviewers' comments:

Reviewer #1 (Remarks to the Author):

This manuscript describes an interesting study of the role of PLC β 2 in inflammatory responses to RNA viruses (enteroviruses) via negative regulation of TAK1 kinase. First of all, PLC β 2 expression was increased in vivo in hand-foot-mouth disease (HFMD) patients and in vitro in CVA16-infected macrophages via TLR3 and p38. Second, mRNA expression of proinflammatory cytokines (TNF, IL-6 and IL-12) were enhanced in the lungs of CVA16-infected Plcb2 $^{-/-}$ mice, and TNF and IL-6 mRNAs and phosphorylation of MAPKs and NF- κ B were increased in CVA16-infected macrophages. Third, physical interactions of PLC β 2 with TAK1 were shown. Fourth, PLC β 2 inhibits TAK1 phosphorylation and activity. An inverse correlation of PLC β 2 expression with TAK1 phosphorylation was found in HFMD patients. Fifth, PLC β 2-mediated inhibition of TAK1 activity is via reduction of PIP2, as PIP2 activates TAK1. Sixth and finally, the enhanced susceptibility to CVA16 virus was found in Plcb2 $^{-/-}$ mice, and on the other hand, PLC activator m-3M3FBS protected mice from CVA16 mice. Overall, this is a novel and well-executed study.

We thank the reviewer for the positive comments about our work.

However, I have issues that preclude my endorsement of the study right away: Nothing has been done on potential redundant functions of other PLC β isoforms. This is particularly important because PIP2 reduction seems a critical factor that regulates TAK1 activity.

We appreciate the reviewer for the insightful comments. PLC β 1-4 control PIP2 turnover both in redundant and specific manner due to distinct tissue distribution and cellular location¹. First, we have examined the interaction of TAK1 with the PLC β 1, PLC β 2 and PLC β 3, and found that only PLC β 2 interact with TAK1 (**Fig. 3h**). Second, we have knocked down the expression of *Plc β 1*, *Plc β 2*, *Plc β 3* and *Plc β 4* by RNAi in primary peritoneal macrophages and found that only knock down of *Plc β 2* markedly promote poly(I:C)-induced *Tnf* and *Il6* expression (**Fig. 2h, i**). Lastly, we found that the mRNA level of *Plc β 2* in macrophages is the most abundant one as compared to other isoforms of PLC β family (**Fig. 2g**). Taken together, these results suggest that PLC β 2 but not other PLC β proteins specifically modulate TAK1 activity.

Along the same line, nothing has been done to address whether direct interaction of PLC β 2 with TAK1 plays any role in the regulation of TAK1 activity.

We appreciate the reviewer for the insightful comments. We have examined the interaction of those PLC β 2 mutants as shown in **Supplementary Fig. 3a, b** with TAK1 and their effect on the activation of TAK1. The data showed that those PLC β 2 mutants that do not interact with TAK1 have no significant inhibitory effect on the TAK1-mediated activation of NF- κ B or AP-1 reporter gene (**Supplementary Fig. 5c**,

d). On the other hand, catalytic domain-deficient mutants or enzyme inactive mutant of PLC β 2 still associated with TAK1, but no longer inhibited the activation of TAK1 signaling (**Supplementary Fig. 3a, b; Supplementary Fig. 5b-d; Fig. 5a-c and Supplementary Fig. 6b**). Together, these results suggest that direct interaction of PLC β 2 with TAK1 is a pre-requisite, but not sufficient for the regulation of TAK1 activity.

Many minor issues:

There is no description on mass spec and no explanation on Fig. 1a.

As per the reviewer's suggestion, we have added more detailed information regarding mass spec in the Figure legends (Line 773); Results (Line 105-109) and Methods section (Line 542-595) of the revised manuscript.

Fig. 3f. There is no baseline control for IP (with non-specific antibody).

As per the reviewer's suggestions, we have included IgG controls in the immuno-precipitation experiments (**Fig 3c**).

Fig. 5. PLC β 2 mutant was not described in sufficient detail. Downstream phenomena used in Fig. 5f, g, o are arbitrarily selected.

According to structure of PLC β 2, histidine located on 327 and 374 are important for the catalytic activity, thus we mutated H327 and H374 to alanine to generate phospholipase-inactive mutant of PLC β 2². As per the reviewer's suggestions, we have

added detail information and related reference in Result (Line 221-223) to describe the PLC β 2 mutant used in **Fig5a-c**.

Thank the reviewer for the comment, we have changed the downstream phenomena in **Fig. 5g** consistently with the one in **Fig. 5f, o**.

Fig. 6a, b. Stats are missing.

As per the reviewer's suggestions, we have added the p value and the stats of the clinical score in **Fig. 6a, b**.

Fig. 7f, g. Stats are missing.

As per the reviewer's suggestions, we have added the p value and the stats of the clinical score in **Fig. 7f, g**.

Line 122. "stimulus" should read "stimuli".

As per the reviewer's suggestions, we have corrected the typo from "stimulus" to "stimuli" (Line 120).

Line 172. "stimulus-dependent". This is not accurate. It should be "constitutive, but affected by poly(I:C) stimulation" or similar.

As per the reviewer's suggestions, we have revised the description of TAK1-PLC β 2 interaction from "stimulus-dependent" to "constitutive, but affected by poly(I:C) stimulation" (Line 179-Line 180).

Line 178. "greatly inhibited" sounds exaggerating.

As per the reviewer's suggestions, we have revised the description from "PLC β 2 greatly inhibited TAK1 phosphorylation" to "PLC β 2 inhibited TAK1 phosphorylation" (Line 190).

Lines 212-213. "wild-type PLC β 2, but not...." This sounds like "all or none" phenomenon.

We thank the reviewer for this suggestion. To avoid a potentially misunderstanding, we revised the description from "wild-type PLC β 2, but not.." to "wild-type PLC β 2 inhibited TAB1 and TAK1 interaction, as well as TAB1-mediated TAK1 phosphorylation, while mutation of phospholipase-active site abolished PLC β 2's inhibition on the interaction of TAK1 with TAB1 and TAK1 phosphorylation" (Line 225-Line 228).

Line 217. "much higher" sounds exaggerating.

We thank the reviewer for this suggestion. To avoid a potentially misunderstanding, we revised the description of our result from "poly(I:C) induced a much higher

phosphorylation of TAK1” to “poly(I:C) induced a more phosphorylation of TAK1”
(Line 241-Line 242).

Lines 229, 236. “a dot blot assay” seems better.

As per the reviewer’s suggestions, we have revised the description from “dotting blot assay” to “dot blot assay” (Line 254 and Line 265).

Line 234. Explain “MODA”.

Per the reviewer’s suggestion, we had provided a detailed description about “MODA” in the results part (Line 258-Line 262).

Line 255. What muscle was analyzed?

We thank the reviewer’s concern. In mice experiments, all the muscle tissue refer to skeletal muscle based on our previous published article³, and we have provided a detailed information in the Result (Line 284; Line 303), Discussion (Line 382) Method (Line 517) and Figure legend (Line 784; Line 849; Line 862) parts.

Line 331. “multiple levels”

As per the reviewer’s suggestions, we have corrected the typo from “multiple level” to “multiple levels” (Line 363).

Line 342. “T and B cells”

As per the reviewer's suggestions, we have revised the typo from "T- and B-cell" to "T and B cells" (Line 373).

Reviewer #2 (Remarks to the Author):

In this study, Wang et al find that RNA virus infection could induce the expression of PLC β 2, which in turns interacts with TAK1 and suppress its activation. The authors then revealed that PIP2, substrate of PLC β 2, is responsible for the activation of TAK1 and subsequent production of pro-inflammatory cytokines in intro and in vivo. This study revealed a novel mechanism of PIP2 in regulating inflammatory response caused by virus infection, which have broad interests to the field of antiviral innate immune response.

We thank the reviewer for the positive comments about our work.

The reviewer has the follower comments:

1. The authors have shown the interaction between PIP2 and TAK1 was essential for TAK1 activation. Moreover, PLC β 2 could disrupt TAK1-TAB1 interaction. is there any link between them?

We thank the reviewer for this suggestive question. Our data demonstrated that PIP2 interacts with and activates TAK1 (**Fig. 5d, f, m**). We have also found that PLC β 2 interact with TAK1 and disrupt the interaction of TAK1 with TAB1 (**Supplementary**

Fig. 4a). PIP2 is shown to dramatically increase the interaction of TAK1 with TAB1 (**Supplementary Fig. 5e**), and PIP2 is degraded by PLC β 2⁴. Furthermore, PLC β 2 is found to interact with and inhibit TAK1 via its phosphodiesterase activity (**Fig. 3** and **Fig. 5a-c**). Therefore, these results suggest that PLC β 2 may disrupt the interaction of TAK1 with TAB1 through hydrolyzing PIP2, thus inhibiting the activation of TAK1.

2. The details about Mass Spectrum analysis should be provided. The experiment was performed with a mixture of different samples or not? The downregulated proteins were not needed in Fig. 1a. The meaning of protein level of “control” and p value in Fig.1b should be clearly stated.

Thanks a lot for the reviewer’s suggestions. As per the reviewer’s suggestion, we have added more detailed information regarding mass spec in the figure legends (Line 773); Results (Line 105-109) and Methods section (Line 542-595).

As per the reviewer’s suggestions, we have deleted those downregulated proteins from **Fig. 1a**. We have also added the stats of the data in **Fig.1b**.

3. Line 127, the authors should explain why they focused on Tlr3 but not other RNA sensors like Tlr7, RIG-I or MDA5. The conclusion about this experiment should be accurate.

As per the reviewer's suggestions, we had revised our title in Figure 6 and Figure 7 to a more accurate and specific description (Line 276 and Line 291). In our previous report, we found that the Tlr3-TRIF signal is crucial for Coxsackievirus A16 (CVA16) infection³. Since PLC β 2-deficient mice had a higher inflammatory response during CVA16 infection and poly(I:C) stimulation (**Fig 4**), we therefore focused on the Tlr3 signal.

To address the reviewer's suggestion, we further investigated the function of PLC β 2 in TLR7 signaling and found that more *I16* mRNA was detected in PLC β 2 knockdown macrophages after TLR7 ligand stimulation (**Supplementary Fig. 2d**) suggesting a negative role of PLC β 2 in the regulation of TLR7 signal. The underlying mechanism awaits for further investigation.

4. The authors showed that Mutant of PLCb2 could not suppress TAK1 activation, how about the binding activity between TAK1 and the mutant? Does the mutant keep the ability to inhibit TAK1-TAB1 interaction?

As per the reviewer's suggestions, we have examined the interaction of phospholipase-inactive mutant PLC β 2 with TAK1, and the effect of this mutant on the TAK1-TAB1 interaction (**Supplementary Fig. 5a, b**). The results indicate that the PLC β 2 enzyme activity is essential for modulating TAK1 activation and TAK1-TAB1 interaction but not the binding of TAK1 with PLC β 2.

5. Line 239, “Tak1^{-/-} A549 cells” should be “TAK1^{-/-} A549 cells”. The expression of TAK1 mutants should be illustrated.

As per the reviewer’s suggestions, we have revised the typo from “*Tak1^{-/-}* A549 cells” to “*TAK1^{-/-}* A549 cells” (Line 268; Line 270; Line 271 and Line 838). We had included the expression of TAK1 mutants in the revised manuscript (**Supplementary Fig. 5f, g**).

6. Line 232, the details of MODA should be provided. Since TAK1 mainly located in cytosol, the authors should explain why TAK1 could bind PIP2 and where the binding occurs.

Per the reviewer’s suggestion, we had provided a detailed description about “MODA” in the results part (Line 258-Line 262).

It is generally thought that TAK1 is mainly located in cytosol, our confocal microscopy analysis revealed that TAK1 co-localize with EEA1 (early endosome marker) in macrophages after poly(I:C) stimulation (**Fig 3g** and **Supplementary Fig. 3c**), suggesting a partial endosomal location of TAK1 in the TLR3 signaling pathway. Since PIP2 is reported to be located on endosome^{1,5,6}, our results suggest that TAK1 could bind with PIP2 at endosome.

7. Line 421, the genetic background of mice should be mentioned.

As per the reviewer's suggestion, we have added background of mice in the Methods section in the revised manuscript (Line 428).

8. Fig. 6 and 7, viral loads data should be provided to confirm whether the pathogenesis were caused by cytokine storm or virus replication.

As per the reviewer's suggestion, we have provided the data to show that the viral loads of CVA16 are comparable in WT and *Plcβ2*^{-/-} mice (**Supplementary Fig. 6a**), suggesting that at least in the CAV16 infections, a more serious pathogenesis of the *Plcβ2*^{-/-} mice may not be caused by virus replication exacerbation.

9. Fig. s6 should be combined into main figure.

As per the reviewer's suggestion, we have replaced the diagram to **Fig 7j**.

Reference

1. Pann-Ghill, Suh., *et al.* Multiple roles of phosphoinositide-specific phospholipase C isozymes. *BMB Rep.* **41**, 415-434 (2008).
2. Hicks, S. N., *et al.* General and Versatile Autoinhibition of PLC Isozymes. *Molecular Cell* **31**, 383-394 (2008).
3. Yang, J., *et al.* Type I Interferons Triggered through the Toll-Like Receptor 3-TRIF Pathway Control Coxsackievirus A16 Infection in Young Mice. *Journal of Virology* **89**, 10860 (2015).
4. Suh, P.G., *et al.* Multiple roles of phosphoinositide-specific phospholipase C isozymes. *BMB reports* **41**, 415-434 (2008).
5. Tan, X., *et al.* Emerging roles of PtdIns(4,5)P₂ - beyond the plasma membrane. *Journal of Cell Science* **128**, 4047-4056 (2015).
6. Yoshida, A., *et al.* Segregation of phosphatidylinositol 4-phosphate and phosphatidylinositol 4,5-bisphosphate into distinct microdomains on the endosome membrane. *Biochim Biophys Acta* **1859**, 1880-1890 (2017).

REVIEWERS' COMMENTS:

Reviewer #2 (Remarks to the Author):

The authors have clarified all the concerns raised during the first round of review.

Reviewer #3 (Remarks to the Author):

I found this manuscript novel, clearly written, and elegantly presented. I feel that all the major concerns raised in the previous round of reviews have been adequately addressed by the authors, including impact of PLCbeta isoforms on TAK1-induced PIP2-mediated regulation of inflammation as well as direct TAK1 interaction with PLCbeta2. The discussion of data obtained from both mouse and human experiments makes this work particularly valuable.

REVIEWERS' COMMENTS:

Reviewer #2 (Remarks to the Author):

The authors have clarified all the concerns raised during the first round of review.

We appreciate the reviewer's pertinent comments.

Reviewer #3 (Remarks to the Author):

I found this manuscript novel, clearly written, and elegantly presented. I feel that all the major concerns raised in the previous round of reviews have been adequately addressed by the authors, including impact of PLCbeta isoforms on TAK1-induced PIP2-mediated regulation of inflammation as well as direct TAK1 interaction with PLCbeta2. The discussion of data obtained from both mouse and human experiments makes this work particularly valuable.

We thank the reviewer for the encouraging comments and for appreciation our efforts in improving the manuscript.